# Cerato-Platanins from Marine Fungi as Effective Protein Biosurfactants and Bioemulsifiers

**DOI:** 10.3390/ijms21082913

**Published:** 2020-04-21

**Authors:** Rossana Pitocchi, Paola Cicatiello, Leila Birolo, Alessandra Piscitelli, Elena Bovio, Giovanna Cristina Varese, Paola Giardina

**Affiliations:** 1Department of Chemical Sciences, University of Naples Federico II, Via Cintia, 80126 Naples, Italy; rosspitocchi@gmail.com (R.P.); p.cicatiello@gmail.com (P.C.); birolo@unina.it (L.B.); apiscite@unina.it (A.P.); 2Department of Life Sciences and Systems Biology, University of Turin, viale P.A. Mattioli 25, 10125 Turin, Italy; elena.bovio@unito.it (E.B.); cristina.varese@unito.it (G.C.V.)

**Keywords:** cerato-platanin, surface tension, foam stabilizer, surface coating, hydrophobin

## Abstract

Two fungal strains, *Aspergillus terreus* MUT 271 and *Trichoderma harzianum* MUT 290, isolated from a Mediterranean marine site chronically pervaded by oil spills, can use crude oil as sole carbon source. Herein, these strains were investigated as producers of biosurfactants, apt to solubilize organic molecules as a preliminary step to metabolize them. Both fungi secreted low molecular weight proteins identified as cerato-platanins, small, conserved, hydrophobic proteins, included among the fungal surface-active proteins. Both proteins were able to stabilize emulsions, and their capacity was comparable to that of other biosurfactant proteins and to commercially available surfactants. Moreover, the cerato-platanin from *T. harzianum* was able to lower the surface tension value to a larger extent than the similar protein from *A. terreus* and other amphiphilic proteins from fungi. Both cerato-platanins were able to make hydrophilic a hydrophobic surface, such as hydrophobins, and to form a stable layer, not removable even after surface washing. To the best of our knowledge, the ability of cerato-platanins to work both as biosurfactant and bioemulsifier is herein demonstrated for the first time.

## 1. Introduction

Marine fungi (MF) have been recognized as a diverse group and an excellent source of natural products. Obligate MF grow and sporulate exclusively in sea water and their spores are capable of germinating in sea water. On the other hand, facultative MF have undergone physiological adaptations to live either in fresh water or terrestrial environment that allow them to grow and possibly also sporulate in the marine environment [1]. MF grow in stressful habitats, such as under cold, lightless and high-pressure conditions or in association with other organisms, sometimes even in oil-spill polluted site. In order to survive in different environmental conditions, they produce many interesting secondary metabolites, with relevant bioactivities, making them attractive in different application fields [2].

Our oceans, seas and coastal zones are under great stress and pollution, particularly because of crude oil, which is a major threat to the sustainability of the planet. However, the presence of crude oil in the environment for millions of years have led to the evolution of many microorganisms able to activate their metabolism to use it as a major or sole source of carbon and energy [3]. The study of these microorganisms is gaining more and more interest in the last decades, in view of developing a completely environmentally friendly bioremediation. In this respect, the Mycotheca Universitatis Taurinensis (MUT) studied the fungal community from a Mediterranean marine site chronically pervaded by oil spills. Some of the fungal isolates were able to use crude oil as their sole carbon source, and *Aspergillus terreus* MUT 271, and *Trichoderma harzianum* MUT 290 were selected because they could represent potential bioremediation agents with strong crude oil degrading capabilities [4].

Many microorganisms have developed mechanism to access hydrocarbons more easily. One way is to increase the bioavailability of these compounds through the production of biosurfactants (BS) [5,6]. BS-producing microorganisms are ubiquitous, inhabiting water (sea, fresh water, groundwater) and land, as well as extreme environments (e.g., oil reservoirs), and thriving at a wide range of temperatures, pH values and salinity. BS offer significant industrial advantages over chemical surfactants thanks to lower toxicity, higher biodegradability, extreme temperatures, varying pH and salinity resistance. They are known to lower the surface and interfacial tension between different phases, possessing a low critical micelle concentration (CMC) and allowing the formation of stable emulsions. Bioemulsifiers (BE) are usually higher in molecular weight than BS as they are complex mixtures of heteropolysaccharides, lipopolysaccharides, lipoproteins and proteins. They can efficiently emulsify immiscible liquids even at low concentrations and thus they possess a good emulsifying activity, but are less effective at surface tension reduction, hence they show a low surface activity [7].

Among the high-molecular weight BS, hydrophobins (HFBs), self-assembling proteins typical of filamentous fungi, are described as the most powerful surface-active proteins. HFBs are small (about 100 amino acids) amphiphilic proteins that play multiple biological roles in fungal biology, lowering the surface tension of the liquid medium in their soluble form and coating aerial structures such as hyphae, fruiting bodies and spores for their easy growth and dispersal in the air, for fungal adhesion to surfaces and host-pathogen interactions [8,9].

Cerato-platanins (CP) are another class of fungal proteins known as surface-active biomolecules [10]. They are small (about 120 amino acids) cysteine-containing proteins (two disulfide bonds), released into the culture filtrate, but also found in the cell wall of fungal hyphae and spores. CP proteins constitute a well-conserved family, with a 70% similarity at some conserved motifs [11]. Similar to HFBs, their solutions lead to strong foam formation and they self-assemble at hydrophobic:hydrophilic interfaces into ordered, amphipathic layers [12,13]. However, they are not HFB-like proteins, and some studies indicated that their behavior is opposite to that of HFBs, i.e., it has been reported that the CP EPL1 from *Trichoderma atroviride* increased the polarity of solutions and surfaces [12,14]. Moreover, structural analysis of CP from *Ceratocystis platani* revealed that there are significant structural differences between CPs and HFBs, i.e., the surface of the molecule shows no large hydrophobic patch [11]. On the other hand, the CP structure (a double ψβ-barrel fold with six β-strands and two α-helices) is homologous to the N-terminal domain of expansins, which are proteins associated with carbohydrate binding and loosening of the cellulose scaffolds in plant cell walls [15]. Expansins are mainly found in plants, where they have various roles in growth and developmental processes, beyond the cell wall-loosening activity. Their action was shown for the first time as a weakening activity on filter paper [16]. Similar to expansins, CPs are able to weaken cellulose substrate, disrupting its non-covalent bonds without any hydrolytic activity. This action could be useful to efficiently hydrolyze cellulosic substrates; in fact, cellulases must have complete access to the cellulose chains that are tightly packed [17]. However, gene knockout experiments have not yet permitted the identification of a clear biological function of CPs, and to date, it is not easy to answer the question of why fungi produce these proteins. CPs can act both as virulence factor and as elicitors. In some plant pathogenic fungi, CPs have been reported to act as phytotoxins inducing cell necrosis. On the other hand, beneficial fungi of the genus *Trichoderma* were shown to induce plant defense responses, activity known as eliciting activity [8,18]. However, the presence and abundant expression of CPs in fungi with all types of lifestyle suggests that the main biological functions are not solely related to fungal–plant interactions but to other, more general aspects of fungal growth.

In this work, we isolated the proteins secreted by *A. terreus* MUT 271 and *T. harzianum* MUT 290 using crude oil as sole carbon source, identified them as CPs (named *At*CP and *Th*CP, respectively) and characterized them for their biosurfactant properties.

## 2. Results

### 2.1. Fungal Growth in Oil as a Carbon Source and Protein Secretion

The marine strains *A. terreus* and *T. harzianum,* isolated from spill-oil polluted site, were grown in the presence of oil as the sole carbon source to induce the production of biosurfactants. Culture broths were depleted of high molecular weight proteins (>30 kDa), by simple ultrafiltration, and then concentrated and dialyzed in 10 mM sodium phosphate buffer pH 7.0. When the proteins secreted by the cultures grown in oil were analyzed by SDS PAGE, only one band was observed in both fungal samples (Figure 1). Otherwise, when the same procedure was applied to the cultural broths of the two fungi grown in rich medium, several protein bands were detected in both cases (Appendix A). Hence, analyses herein reported were performed on the homogeneous samples obtained from the oil supplemented cultures with the abovementioned procedures. The amount of proteins secreted as function of the salt concentration in the cultural broth was evaluated (Appendix A of Appendix A), and the maximum protein yields (2.7 ± 0.3 and 1.7 ± 0.5 mg/L of cultural broth of *A. terreus* and *T. harzianum*, respectively) were obtained at 30 g/L of NaCl. At least three biological replicated were analyzed in each condition.

The proteins obtained in these conditions from the marine strains of *T. harzianum* and *A. terreus* were named *Th*CP and *At*CP, respectively.

### 2.2. Emulsification Tests

To investigate the performances of these proteins as emulsifying agents, emulsification tests were performed using Dectol (a mix of Decane-Toluene 65:35, *v/v*), according the protocol of Blesic et al. [19]. The tests were carried out at two protein concentrations, 0.05 and 0.1 mg/mL. As negative control, a sample containing only phosphate buffer was used.

*At*CP showed an emulsification index E_24_ value slightly higher than *Th*CP (Figure 2) at the tested concentrations.

The protein nature of these BS was confirmed by incubation of the samples with proteinase K, since emulsification activity was quite lost after this treatment (Appendix A).

### 2.3. Protein Identification and Characterization

A proteomic approach was adopted to identify the isolated proteins. Only one protein band was visible in each of the SDS PAGE, and those bands were excised and submitted to a classical proteomic approach to identify the proteins. Only a single protein from *T. harzianum* and one from *A. terreus* could be confidently identified in the respective protein bands; therefore, these samples can be considered homogeneous, at least according to the commonly accepted criteria of protein purification assessment. The identified proteins are ascribed to the *Cerato Platanins* family (Table 1), classified as a group of extra-cellular, small cysteine-rich fungal proteins. This family is characterized by high hydrophobicity like HFBs. In addition, cysteines and tryptophans are reported to be essential for their functions, with the four conserved cysteines involved in the formation of two disulphide bridges and the tryptophan important for their eliciting ability [20].

Analysis of the circular dichroism (CD) spectra (Figure 3) showed that the two proteins exhibit different conformations. While the α-helix was the preponderant secondary structure in *At*CP (38% α-helix, 16% β-strands, 18% turns, 28% random coil), a higher percentage of β-structure (17% α-helix, 30% β-strands, 20% turns, 33% random coil) was observed in the case of *Th*CP. It is worth to note that the protein samples stored at 4 °C retained these conformations for at least one month.

### 2.4. Surface Tension Measurements

The protein samples were diluted with 10 mM phosphate buffer, pH 7, at different concentrations, starting from the maximum concentration achievable, up to the *γ* value of the buffer used as reference. Indeed, at protein concentrations of *Th*CP higher than 0.10 mg/mL (7 × 10^−6^ M) and of *At*CP higher than 0.17 mg/mL (1 × 10^−5^ M), aggregation phenomena occur. In Figure 4, the surface tension trends as function of protein concentration (M) are shown.

A very low value of *γ* (36.4 mN/m) was observed in the case of *Th*CP with respect to that of *At*CP (56.3 mN/m). The critical micelle concentration (CMC), which indicates the minimum concentration of biosurfactant necessary to achieve the lowest stable surface tension, was 8 × 10^−7^ M (0.01 mg/mL) in the case of *Th*CP, significantly lower than the CMC of *At*CP (9 × 10^−6^ M, 0.12 mg/mL).

### 2.5. Cellulose Loosening

We tested the ability of both proteins to weaken cellulose, according to the protocol set up by Baccelli et al. [16]. The action of CPs on cellulose was clearly observable as turbidity increases because paper fragments are released from filter paper discs. In order to quantify solution turbidity, the absorbance at 500 nm of each sample was measured after three days of incubation at 37 °C. A solution containing bovine serum albumin (BSA) was used as control. Hence, *At*CP and *Th*CP were both able to weaken cellulose (Figure 5).

To verify if the action of CPs on paper improved the availability of cellulose in solution, the amount of reducing sugars was measured in the supernatants of the CP samples incubated with paper. The increase of cellulose availability was measured using the DNS method. Figure 5 shows the amount of reducing sugars detected before and after cellulase treatment for both proteins. Both CPs were capable to triple the quantity of reducing sugars in solution, compared to the control.

### 2.6. Water Contact Angle (WCA) Measurement

CPs were deposited on hydrophobic surfaces (Teflon) and hydrophilic surfaces (glass) and WCA was analyzed after drying and phosphate buffer washing. The deposition on glass surface did not reveal significant changes of surface wettability. On the opposite, interesting results were obtained using the hydrophobic surface. When deposited on Teflon, both *Th*CP and *At*CP layers increased the wettability of the surface, as shown by the alteration of the water droplet shapes (Figure 6). The treatment of the plastic material was also resistant to extensive phosphate buffer washing.

## 3. Discussion

In the search for new protein biosurfactants with improved performances with respect to the already known ones, the proteins secreted by marine fungi isolated in polluted sea by oil spills were analyzed. In particular, the strains *T. harzianum* MUT 290 and *A. terreus* MUT 271 [4], able to grow in the presence of crude oil as their sole carbon source were examined for protein biosurfactant production since this condition should induce the production of biosurfactants to solubilize organic molecules, as a necessary step in the metabolizing process of lipophilic molecules [6].

Previously, protocols were set up to isolate the most surface-active known proteins, the HFBs, from fungal culture broth [21]. When these protocols were herein used with culture broths of *T. harzianum* and *A. terreus*, only one secreted protein band could be detected by SDS PAGE in both cases, in which only one protein could be confidently identified. Then, a very simple protocol was adopted to obtain homogeneous samples of these proteins.

Generally speaking, high molecular weight BS, such as proteins, can be better defined as BE, being able to stabilize emulsions more than to reduce the surface tension of water [7,22]. Therefore, as a first approach, the ability of both proteins to stabilize emulsions was verified. Indeed, the obtained E_24_ values were comparable to those of other proteins and to commercially available surfactants analyzed in the same way [19,23].

Both these proteins were identified and ascribed to the family of cerato-platanins (CP): small, conserved, hydrophobic proteins, whose function is still a matter of debate. Even if CPs have been included among the fungal surface-active and surfactant proteins [10], to the best of our knowledge, no clear evidence of these activities has been reported thus far. As a matter of fact, Frischmann et al. [12] reported that the cultural broth of *T. atroviride* containing CP proteins produced a lot of stable foam, but no other analytical study on this CP characteristic was performed. We also analyzed the surface tension reduction of these CPs and unexpectedly observed that *Th*CP behaves as a good BS, beyond being a good BE. Indeed, the surface tension value reached in the presence of *Th*CP was much lower than those of *At*CP and of Sap-*Pc*, another BE protein recently purified from a marine strain of *Penicillium chrysogenum*, measured in comparable conditions [23].

Another issue concerned the ability of CPs to change the wettability of surfaces and the comparison of their behavior to that of HFBs. At first, CPs were described as a class of proteins like that of HFBs [24]; afterwards, Frishmann et al. [12] reported that CPs increased the polarity of surfaces rather than decreased it, differently from HFBs, whose layers inverted the wettability of surfaces. We analyzed the behavior of the two CPs using the same procedure adopted for HFBs [21] and observed that both CPs were able to make hydrophilic a hydrophobic surface, like the HFBs. They also formed a stable layer, which was not removed even after washing of the surface. The similar properties of the two classes of proteins, CPs and HFBs, are related to their common BS abilities, which should be the result of the presence of an exposed hydrophobic patch on the surface of the molecules, rendering them amphiphilic. However, while this characteristic was observed in the known three-dimensional structures of HFBs, it was not an evident feature of the structure of CP from *Ceratocystis platani* [15], and in the deposited structure of Sm1 (3M3G in Protein Data Bank) from *Trichoderma virens* [13], a protein whose sequence is 92.5% identical to SnodProt1, the protein we identified in *T. harzianum,* herein named *Th*CP. However, the exposure of a flexible and hydrophobic moiety of the protein molecule could also occur in the presence of hydrophobic/hydrophilic interfaces, and this could eventually be an alternative and plausible mechanism to interact with hydrophobic surfaces. It is also worth noting that the CD spectrum of *Th*CP is different from those reported for other CPs [25,26], thus suggesting that the *Th*CP structure is indeed consistently different from the other cases reported, at least in the conditions we tested.

As far as the comparison of CPs with HFBs is concerned, it was reported that some CPs form amyloid-like aggregates, but only under stress conditions [26,27]. In fact, *Th*CP and *At*CP do not show the same propensity to form amyloid-like fibrils as Class I HFBs, since the CD spectra of *Th*CP and *At*CP were unchanged even after one month, differently from what occurs to Class I HFBs, which change their conformation after some days in mild conditions [28,29].

An extremely interesting feature of CPs is their ability to weaken cellulose, similar to another class of fungal proteins, the expansins [16]. We observed a remarkable cellulose loosening activity of both CPs, which allowed a much more effective attack of cellulase on its substrate, disrupting the ordered chains of cellulose and increasing its surface area. This feature allows to envisage other potential biotechnology applications of CPs, in addition to those related to their BS activity, such as, for example, in the pre-treatment of lignocellulose waste materials in bioconversion processes to biofuels.

In conclusion, we identified two CPs, secreted by marine fungi grown on oil as sole carbon source, and characterized them for their BE property. Notably, *Th*CP showed very good performance in reducing surface tension, thus possessing both surfactant and emulsifying activities, which can suggest, as with other BS and BE, its exploitation in a broad range of industrial application fields, such as medical, pharmaceuticals, food and beverages, cosmetics, agriculture, agrochemical, paint, detergent, textiles and petrochemical production.

## 4. Materials and Methods

### 4.1. Fungal Growth

MF strains were maintained through periodic transfer on agar plate at 20 °C, using XNST30 agar medium (malt extract 3 g/L; agar 15 g/L; yeast extract 3 g/L; NaCl 30 g/L; 10 g/L glucose and 5 g/L peptone) [21].

Mycelium disks (10 mm diameter) were taken from the margin of the actively growing colonies, and pre-inoculated in 100 mL flasks containing 50 mL of ONR7 mineral medium [30] supplemented with 10 g/L glucose, 2 g/L peptone, in triplicates and incubated at 28 °C for 5 days. Then, 50 mL of these pre-inoculum were inoculated in 1 L flasks containing ONR7 (500 mL) added with 1% *v/v* lampant oil and 0.1% *v/v* Tween 80. Lampant oil is a clear oil used to prevent corrosion to protect mechanical compounds of industrial and laboratory machines. It contains a rich hydrocarbon fraction that may be a suitable substrate for biosurfactant production. Flasks were inoculated in triplicates and incubated in the dark at 28 °C for 14 days.

### 4.2. Purification of the Proteins

Mycelia were separated from the culture broth using a Whatman 3MM paper filters (Termo Fischer Scientific, Rodano (MI), Italy). Culture broth was filtered using Stericup GP vacuum filtration system (0.22 μm pore size, Merck KGaA, Darmstadt, Germany), and biosurfactants were concentrated by air bubbling, using a Waring blender. The formed foam was separated and collected until further foam was not formed. The collected foam was loaded in an Amicon Ultrafiltration cell equipped with a 30 kDa cut-off PES Millipore Ultrafiltration Disc (Merck KGaA, Darmstadt, Germany). The ultrafiltrate, depleted of high molecular weight proteins (>30 kDa), was concentrated using the same cell equipped with a 3 kDa cut-off disc and dialyzed in 10 mM sodium phosphate buffer pH 7.0. Protein concentration was evaluated using the Pierce 660 nm Protein Assay kit (Termo Fischer Scientific, Rodano (MI), Italy) using bovine serum albumin as standard.

### 4.3. Emulsification Activity

The emulsification capability of samples was investigated at different protein concentrations. In the experiment, 2 mL of Dectol (a mix of Decane-Toluene 65:35, *v/v*) was added as emulsifying agent to 1 mL of each surfactant protein in 10 mM phosphate buffer (pH 7.0) in 5 mL glass vials [19]. This mixture was homogenized using the IKA T-10 Basic Ultra Turrax Homogenizer IKA-Werke GmbH, Staufen, Germany) for 2 min; then, the stability of the emulsions after 24 h was evaluated. The stability is reported in terms of emulsification index E_24_:

$$E_{24}=\frac{height\;emulsion}{total\;height}\times 100$$ Proteins samples, 0.05 mg/mL, were incubated with Proteinase K (from *Tritirachium album*, Sigma-Aldrich, St. Louis, MO, USA) 1 mg/mL at 37 °C for 10 min. Then, an emulsification test was performed as described above.

### 4.4. Protein Identification by Mass Spectrometry

Protein identification was performed on Coomassie Blue stained bands excised from mono-dimensional SDS PAGE (15%). For each sample, 3 μg of proteins were treated with sample buffer containing 0.1 M DTT, and then, were further denatured, boiling them at 100 °C for 10 min.

Because these proteins are poorly stained with Coomassie Brilliant Blue, the samples were loaded in duplicates on the same gel and a set of samples was Coomassie stained, whereas the other set was analyzed by silver staining. By superimposing the Coomassie-stained and silver-stained protein bands, we localized the protein bands corresponding to the proteins of interest (apparent molecular mass between 12 and 17 kDa), which were excised from gel and processed as reported in Cicatiello et al. [23] on a 6520 Accurate-Mass Q-TOF LC/MS system (Agilent Technologies, Palo Alto, CA, USA) equipped with a 1200 HPLC system and a chip cube. The acquired MS/MS spectra were transformed in Mascot Generic Format (.mgf) and used to query the NCBI nonredundant (NR) database (http://www.ncbi.nlm.nih.gov), version 20160114, 79 354 501 sequences and 28 992 349 963 residues for “Fungi” as taxonomy restriction, for protein identification with a licensed version of MASCOT software (www.matrixscience.com) version 2.4.0, which was performed as previously described [25]. Only proteins presenting two or more peptides, with at least one above the ion score threshold of 40, were considered as positively identified. the ion score was −10 log(P), where P is the probability that the observed match is a random event. Individual ion scores >40 indicated the identity or extensive homology (*p* < 0.05).

### 4.5. Circular Dichroism (CD) Spectroscopy

CD spectra were recorded on a Jasco J715 spectropolarimeter (Jasco Corporation, Cremella (LC), Italy) equipped with a Peltier thermostatic cell holder in a quartz cell (0.1 cm light path) from 190 to 250 nm. The temperature was kept at 20 °C, and the sample compartment was continuously flushed with nitrogen gas. The final spectra were obtained by averaging three scans, using a bandwidth of 1 nm, a step width of 0.5 nm and a 4 s averaging per point. Secondary structure analysis of the collected CD data were processed by DichroWeb [31], using Selcon3 and ContiLL algorithms.

### 4.6. Surface Tension Measurement

The surface tension, γ, of the extracted proteins was measured with a Sigma 70 tensiometer (KSV, Stockholm, Sweden) using the Du Noüy ring method as described elsewhere [32]. The ring is submerged into the solution and then slowly pulled through the liquid–air interface to detach it from the interface, *γ* was correlated with the force required to raise the ring from the surface of the liquid–air interface. Aliquots of the proteins suspended in phosphate buffer pH 7.0 were prepared at different concentrations; each sample was filtered with a 0.22 μm filter before testing. In total, 7 mL of sample was added to the vessel, mixed using a magnetic stirrer and allowed to equilibrate 5 min prior to measuring the surface tension. Phosphate buffer 10 mM with surface tension of 72 mN/m was used to calibrate the tensiometer.

### 4.7. Cellulose Loosening and Cellulolytic Activity Assay

Whatman filter paper (3MM) was used as substrate to assay the CPs effect on cellulosic materials [16]. Five mm diameter paper disks were incubated in 1 mL of 10 mM phosphate buffer, pH 7.0, containing 0.05 mg/mL of CPs from *T. harzianum* and *A. terreus*. Bovine serum albumin (BSA) was used as negative control. The experiments were performed in Pyrex 20 mL glass culture tubes and samples incubated at 37 °C, for 72 h, onto an orbital shaker at 320 rpm. At the end of the incubation period, disks were removed and the absorbance at 500 nm was measured, in order to quantify the turbidity of the obtained dispersion.

The presence of reducing sugars (RS) was verified using the dinitrosalicylic acid (DNS) method. The obtained dispersions were centrifuged (5 min at 5000 rpm). In total, 300 μL were directly tested with DNS; 300 μL was further incubated for 3 h at 37 °C, under agitation, 320 rpm, in the presence of 1 U of cellulase (≥0.3 units/mg solid, Sigma-Aldrich, St. Louis, MO, USA). After the addition of 300 μL of 1% DNS solution, all samples were incubated at 90 °C for 15 min, cooled at room temperature and the absorbance at 575 nm was then measured. The absorbance values were interpolated into a glucose calibration curve made in 10 mM sodium phosphate buffer (pH 7.0) to determine the concentration of reducing sugars. The reducing sugars released were then measured by the DNS method as previously described.

### 4.8. Water Contact Angle Determination

To test the ability of the proteins to self-assemble into a stable amphiphilic layer and to functionalize a solid surface, protein samples (50 μL of 0.05 mg/mL protein solution) were deposited on politetrafluoroetilene (PTFE) or glass, dried at 60 °C and washed with phosphate buffer. Drop shapes were modeled with the software program Image J (Rasband, W.S., ImageJ, U. S. National Institutes of Health, Bethesda, Maryland, USA, https://imagej.nih.gov/ij/, 1997-2018).

## Figures and Tables

**Figure 1 ijms-21-02913-f001:**
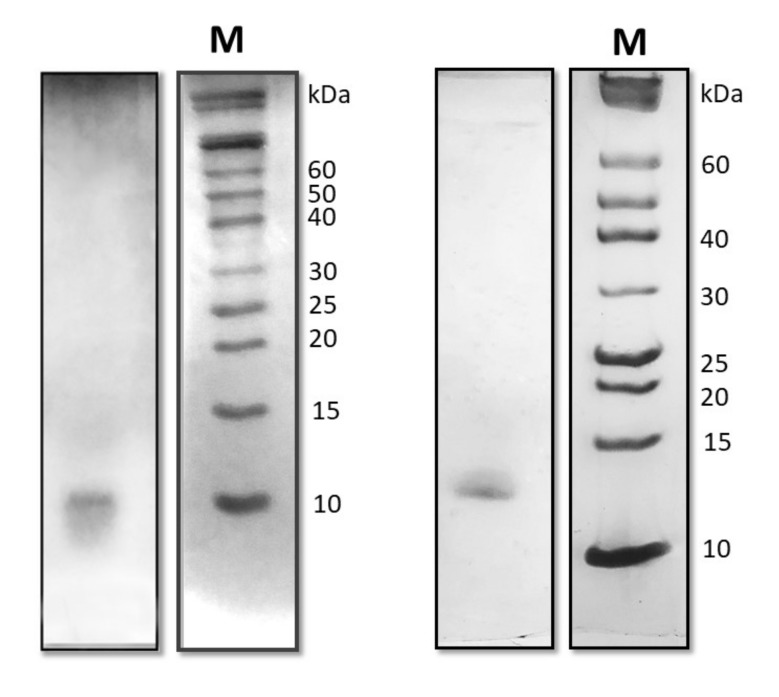
Silver stained SDS PAGE of proteins secreted by *T. harzianum* (left) and *A. terreus* (right) grown in the presence of 1% lampant oil as carbon source and 30 g/L NaCl (M: molecular weight markers).

**Figure 2 ijms-21-02913-f002:**
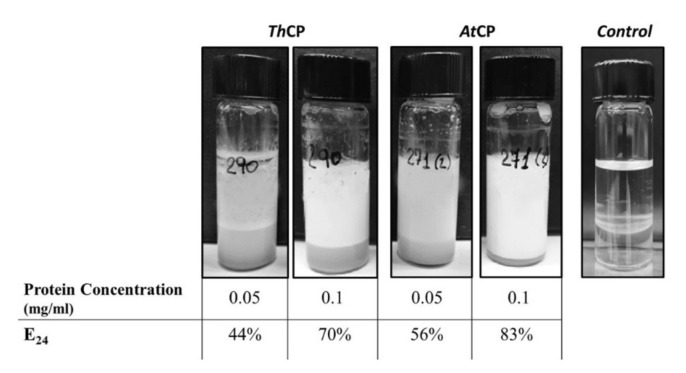
Dectol emulsions at two different protein concentrations. *Th*CP and *At*CP solutions were mixed with Dectol (2:1 *v/v*) and analyzed after 24 h. Dectol emulsion with 10 mM Phosphate buffer, pH 7, is reported as control.

**Figure 3 ijms-21-02913-f003:**
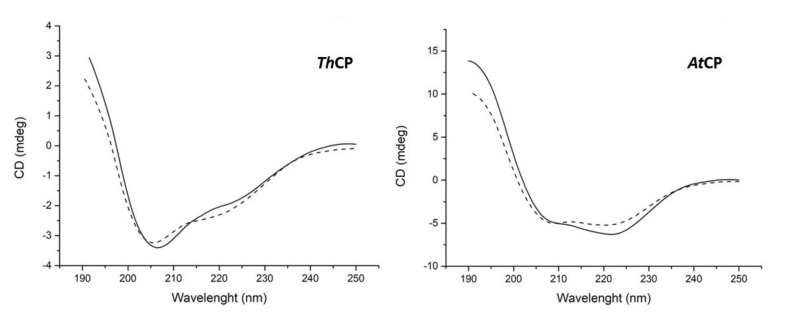
Circular dichroism (CD) spectra of CPs (0.05 mg/mL) from *T. harzianum* (*Th*CP) and from *A. terreus* (*At*CP) dissolved in 10 mM phosphate buffer pH 7. Dotted lines indicate CD spectra of the same samples acquired after 1 month of storage.

**Figure 4 ijms-21-02913-f004:**
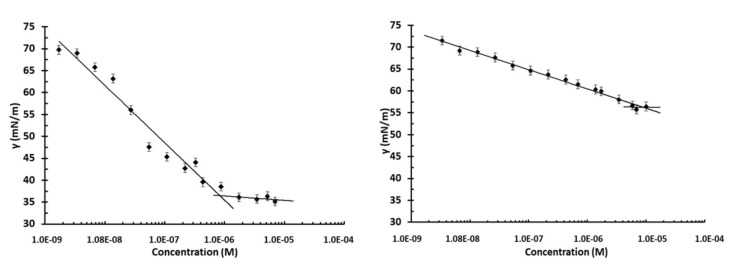
Surface tension of *Th*CP (left) and *At*CP (right) in 10 mM phosphate buffer pH 7, as a function of protein concentration.

**Figure 5 ijms-21-02913-f005:**
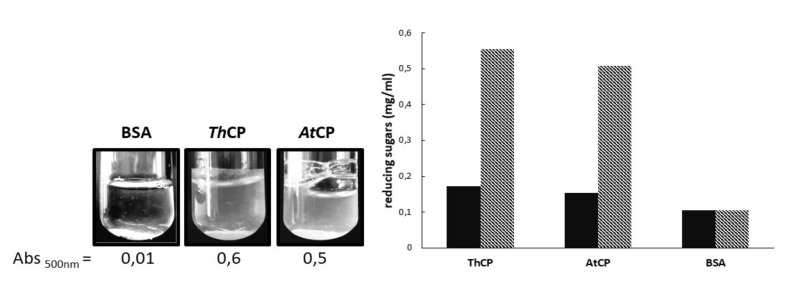
Left panel: protein samples (0.05 mg/mL), after incubation for three days at 37 °C in the presence of a paper disc, and the corresponding absorbance values at 500 nm. Right panel: concentration of reducing sugars (RS) measured for each sample. Continuous black columns: RS determined on the supernatants of the samples described above. Dashed columns: RS determined on the samples described above, further incubated with cellulase (1 U tot). 0.05 mg/mL. Bovine serum albumin (BSA) was used as reference.

**Figure 6 ijms-21-02913-f006:**
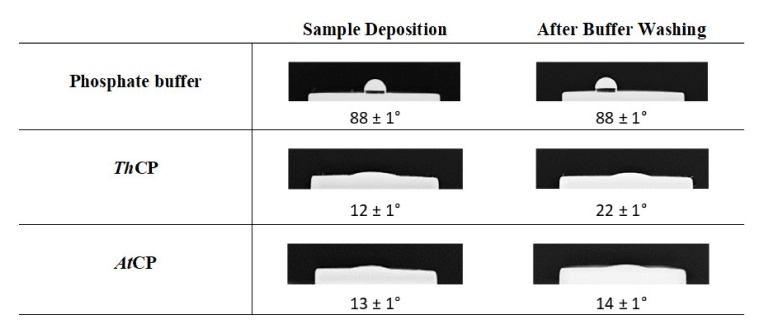
Water Contact Angle (WCA) analysis of hydrophobic surface (PTFE) upon deposition of CPs, before and after washing with 10 mM Phosphate buffer.

**Table 1 ijms-21-02913-t001:** Proteins were identified by searching NCBI database with MSMS Ion search Mascot software (Matrix Science) with Fungi as the taxonomy restriction, with carboxyamidomethylation of Cys as fixed modifications, and oxidation on Met, pyro-Glu formation at Gln at the N-terminus of peptides as variable modifications.

Sample	Protein name^a,b^ (Entry Code)	Family	Peptides (Ion Score)	Sequence Coverage (Number of Peptides)
*T. harzianum*	*Th*CP^a^ SnodProt1^b^ (gi|818166392)	Cerato-platanins	R.YHWSTQGQIPR.F (25)	45% (4)
			R.SLNVVSCSDGPNGLETR.Y (70)	
			R.FPYIGGVQAVAGWNSASCGTCWK.L (28)	
			R.VSATASQVAVK.N (57)	
*A. terreus*	*At*CP^a^ allergen Asp f 15^b^ (gi|115384120)	Cerato-platanins	K.LTYGGK.S (29)	31% (4)
			K.WPTFGSVPK.F (26)	
			R.VQATYQEVAK.S (54)	
			K.FPHIGGSPTIPGWNSPNCGK.C (35)	

Only proteins presenting two or more peptides, with at least one above the ion score threshold of 40 were considered as positively identified. Ion score was −10 log(P), where P is the probability that the observed match is a random event. Individual ion scores >40 indicated identity or extensive homology (*p* < 0.05). Peptides are reported with the flanking residues with dots to indicate tryptic cleavage sites. ^a^name of the protein, herein attributed; ^b^name of the identified protein.

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
