# Peer review of "Cerato-Platanins from Marine Fungi as Effective Protein Biosurfactants and Bioemulsifiers"

_ijms, 2020, doi:10.3390/ijms21082913_

Round 1
Reviewer 1 Report
In the manuscript „Protein biosurfactant from marine fungi” by Pitocchi and colleagues the authors investigate two marine fungi for secretion of proteinaceous biosurfactants during growth on crude oil as sole carbon source. The crude oil is supposed to induce the formation of biosurfactants needed for enhancing the bioavailability of this C-source.
The authors identified two novel proteins of the cerato-platatin family in the culture supernatant of the two strains. These proteins were mainly related to plant virulence until now. Here, the authors purified the two proteins and characterized them by common tests for surfactants and emulsifiers. The results suggest that both proteins have both properties, a new finding of the field. Thus, they might represent promising compounds for industrial use. In addition, the authors find a cellulose-loosening ability, which is known from other cerato-platatins in filamentous fungi.
While the finding of the manuscript itself is in principle interesting for the biosurfactant/bioremediation community I do not favour publication in the current state. The major weakness is that the manuscript and especially the results section are not very clearly written and not elaborate enough, the results and used methods are not very well explained and some mentioned results not shown.
Major comments:
- I missed a description of the purification procedure and of the purity of the obtained proteins.
- The proteins were not clearly defined in terms of their names (SnodProt1 and Aspf15) in the text after their identification (only in the figure legend) (paragraph lines 119-125, p.4)
- The significance of the CD spectroscopic results is not clear to me from the text (what do we learn from the results?).
- The purification strategy was not mentioned in the results section, however, this is important to follow the rest of the experiments. Which purity was obtained and how much protein was obtained from 1 L of culture?
- To enhance the significance of the whole manuscript, an experimental approach showing that the substances indeed enhance the bioavailability of crude oil would be desirable.
Minor comments:
- The manuscript header is very general and not nicely explaining the major finding of the paper
- Figure 2: It should read phosphate buffer instead of phosphate buffer in the heading for the control experiment
- Please show results for salt concentration experiments (p3, line 100 ff) in the supplementary
- Please define “high molecular” in the results text (p3, line 97)
- Please show results for proteinase K treatment (p4, line 116f)
Author Response
Major comments:
- I missed a description of the purification procedure and of the purity of the obtained proteins.
We agree that the purification procedure was not clearly described in the previous version of the manuscript. In this new version, both Materials and Methods and Results sections have been modified to clarify this aspect (lines 98-111,275-281). The purity of the proteins was assessed by SDS PAGE.We avoided classical purification techniques, reducing to a minimum the purification steps, on the basis of our experience in handling hydrophobins. Indeed, the purification yield of any chromatography step can be very low for these peculiar proteins, due to their marked adhesion properties on the matrixes. However, in a first approach we tried a more complex purification procedure that we had set up for hydrophobins (see ref.20), obtaining the same results in term of protein purification, but those proteins were eventually fully denatured, as we ascertained by CD spectroscopy. Therefore, we developed and adopted the very simple procedure described in the manuscript.
- The proteins were not clearly defined in terms of their names (SnodProt1 and Aspf15) in the text after their identification (only in the figure legend) (paragraph lines 119-125, p.4)
We modified Table 1, and added a sentence to clarify the protein names (lines 110-111), and these new names have been used throughout the manuscript.
- The significance of the CD spectroscopic results is not clear to me from the text (what do we learn from the results?).
Actually, the significance of the CD spectra reported was not explained in the previous version of the manuscript. In the new version we have changed the figure and given some details on the significance of recording these CD spectra in the Discussion section, comparing them to others reported in literature (lines 240-247).
- The purification strategy was not mentioned in the results section, however, this is important to follow the rest of the experiments. Which purity was obtained and how much protein was obtained from 1 L of culture?
As mentioned above, we have also modified the first paragraph of the Results section. The protein yields were indicated, and a table has been added in Supplementary Materials, reporting the protein yields at the different NaCl concentrations tested.
- To enhance the significance of the whole manuscript, an experimental approach showing that the substances indeed enhance the bioavailability of crude oil would be desirable.
This approach would be very interesting; however, in our opinion it concerns aspects which are beyond the aim of this manuscript, focused on the characterization of the protein activities more than on physiological aspects of the fungal growth.
Minor comments:
- The manuscript header is very general and not nicely explaining the major finding of the paper
The header has been modified
- Figure 2: It should read phosphate buffer instead of phosphate buffer in the heading for the control experiment
The heading of the figure has been modified
- Please show results for salt concentration experiments (p3, line 100 ff) in the supplementary
As mentioned above, a table has been added in the Supplementary Materials showing these results
- Please define “high molecular” in the results text (p3, line 97)
The definition has been added. It is related to the ultrafiltration membrane cut off, 30kDa.
- Please show results for proteinase K treatment (p4, line 116f)
A figure showing these results has been added in Supplementary Materials
Reviewer 2 Report
Pitocchi and Co-authors describe the isolation of small amphiphilic proteins secreted by two oil degrading marine fungi, and characterized them with regard to their surfactant properties and the weakening of cellulose structures. The manuscript is overall concise and well written. Nonetheless I would like to suggest some minor modification which may help to increase to impact of the study and the merit for the scientific community.
l. 95 The authors line-out during the introduction that CP-family protein can fulfill a wide range of functions for the producing fungi. Although a function as emulsifying agent for marine oil degraders is supported by the literature (e.g. Mar Drugs 2019, 17:408) and I understand completely that it was not within the scope of this study to investigate the physiological background of CPs production, it would be very interesting to know if the authors conducted cultures without oil supplementation and if surface tension reduction/CP production was likewise observed therein (or in the pre-cultures). If this was not conducted, it would be nice to see discussed how the authors think about a strict alkane dependence of CP production in these strains.
l. 102. In accordance with the journal guidelines, I suggest to include the results on media optimization into the publication.
l. 126/table1. Is it likely that the isolated proteins are completely identical with the reference sequence despite the different strain background? Otherwise, I suggest to amplify and sequence the respective genes from the genomes of the MUT strains and deposit their sequences in a public repository.
l. 218. I miss some comments on the relevance of the CDspec data for this study. I think the authors may discuss it here in the context of the known structural features of the similar protein and compare its secondary structure composition with their CD results.
l. 230 ff. Since application -specific characterization was not in the scope of this study and it rather focused on general surface active properties, I think the conclusion should be weakened a bit adding something like "like other BE or biosurfactants" to illustrate the current state of knowledge.
Author Response
- 95 The authors line-out during the introduction that CP-family protein can fulfill a wide range of functions for the producing fungi. Although a function as emulsifying agent for marine oil degraders is supported by the literature (e.g. Mar Drugs 2019, 17:408) and I understand completely that it was not within the scope of this study to investigate the physiological background of CPs production, it would be very interesting to know if the authors conducted cultures without oil supplementation and if surface tension reduction/CP production was likewise observed therein (or in the pre-cultures). If this was not conducted, it would be nice to see discussed how the authors think about a strict alkane dependence of CP production in these strains.
We made cultures in rich media, without oil, and observed different protein patterns. These results have been included in this new version of the manuscript and a figure has been added in the Supplementary Materials. However, this issue is very interesting and deserves a deeper analysis, which will be faced in the future, also performing RT-PCR experiments to determine the difference in the transcription levels of these genes in the different culture conditions.
- 102. In accordance with the journal guidelines, I suggest to include the results on media optimization into the publication.
These results have been added, as a table in Supplementary Materials.
- 126/table1. Is it likely that the isolated proteins are completely identical with the reference sequence despite the different strain background? Otherwise, I suggest to amplify and sequence the respective genes from the genomes of the MUT strains and deposit their sequences in a public repository.
This issue is also interesting, we cannot exclude that some differences in the sequences of our proteins with respect to the reference sequences can occur, albeit it must be highlighted that cerato-platanins are very conserved proteins. This analysis will be the subject of a future work, together with the transcriptional analysis.
- 218. I miss some comments on the relevance of the CDspec data for this study. I think the authors may discuss it here in the context of the known structural features of the similar protein and compare its secondary structure composition with their CD results.
Actually, the significance of the CD spectra reported was not explained in the previous version of the manuscript. In the new version we have changed the figure and given some details on the significance of recording these CD spectra in the Discussion section, comparing them to others reported in literature (lines 240-247).
- 230 ff. Since application -specific characterization was not in the scope of this study and it rather focused on general surface active properties, I think the conclusion should be weakened a bit adding something like "like other BE or biosurfactants" to illustrate the current state of knowledge.
Modifications have been made.
Round 2
Reviewer 1 Report
The authors addressed all points raised by the two reviewers in their revised version. In my opinion, the manuscript is now suited for publication in IJMS.
Author Response
Thank you very much for your valuable comments.